# Data-Driven Simulation for Evaluating the Impact of Lower Arrival Aircraft Separation on Available Airspace and Runway Capacity at Tokyo International Airport

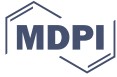

**Katsuhiro Sekine** [1] , **Furuto Kato** [2] , **Kota Kageyama** [3] **and Eri Itoh** [2,3,*]

1    Department of Industrial Management and Engineering, Tokyo University of Science, Tokyo 125-8585, Japan; 4419519@ed.tus.ac.jp

2    Department of Aeronautics and Astronautics, The University of Tokyo, Tokyo 113-8656, Japan; furuto787@g.ecc.u-tokyo.ac.jp

3    Air Traffic Management Department, Electronic Navigation Research Institute, Tokyo 182-0012, Japan; kage@mpat.go.jp

\*    Correspondence: eriitoh@g.ecc.u-tokyo.ac.jp

**Abstract:** Although the application of new wake turbulence categories, the so-called "RECAT (wake turbulence category re-categorization)", will realize lower aircraft separation minima and directly increase runway throughput, the impacts of increasing arrival traffic on the surrounding airspace and arrival traffic flow as a whole have not yet been discussed. This paper proposes a data-driven simulation approach and evaluates the effectiveness of the lower aircraft separation in the arrival traffic at the target airport. The maximum runway capacity was clarified using statistics on aircraft types, stochastic distributions of inter-aircraft time and runway occupancy time, and the levels of the automation systems that supported air traffic controllers' separation work. Based on the estimated available runway capacity, simulation models were proposed by analyzing actual radar track and flight plan data during the 6 months between September 2019 and February 2020, under actual operational constraints and weather conditions. The simulation results showed that the application of RECAT would reduce vectoring time in the terminal area by 7% to 10% under the current airspace and runway capacity when following a first-come first-served arrival sequence. In addition, increasing airspace capacity by 10% in the terminal area could dramatically reduce en-route and takeoff delay times while keeping vectoring time the same as under the current operation in the terminal area. These findings clarified that applying RECAT would contribute to mitigating air traffic congestion close to the airport, and to reducing delay times in arrival traffic as a whole while increasing runway throughput. The simulation results demonstrated the relevance of the theoretical results given by queue-based approaches in the authors' past studies.

**Keywords:** air traffic management; data-driven approach; modeling and simulation; RECAT (wake turbulence category re-categorization)

## 1. Introduction

Increasing the runway throughput is one of the most promising solutions to accommodate future air traffic growth. In this regard, several applications that lower the aircraft separation have been discussed. New wake turbulence categories, the so-called "RECAT" (wake turbulence category re-categorization) [1–3], reduce aircraft separation minima according to the combination of the aircraft types of the succeeding and preceding aircraft at the target airport. To ensure operational safety, the separation margin is given in addition to the separation minima [4]. This separation margin is determined by the levels of the automation systems which support the work of the air traffic controllers (ATCOs) and pilots on the ground and in the air. Consequently, applying RECAT will affect aircraft arrival flows in that the reduced separation will lead to lower Inter-Arrival Time (IAT), particularly with the increased air traffic volumes in the future.

Several studies have targeted efficient use of runway capacity at the case study airports. Bubalo et al. (2011) [5] investigated the impact of increased demand at Berlin Brandenburg International Airport on runway capacity through simulations using an airport and airspace modeling tool, SIMMOD [6]. This simulator outputs data to calculate the capacity utilization and selected service levels by showing the average delay time per flight according to input scenario including flight schedules. They showed that the capacity of 76 flights per hour was the maximum demand provided under defined assumptions. Irvine et al. (2015) [7] quantified and examined possible relative capacity gains across London by conducting Monte Carlo simulations with mathematical probability model. In their model, the Runway Occupancy Time (ROT) used in the past by the authors' group study [8] was applied by taking over the ICAO rule, and a fixed value was used for each wake turbulence category. They found that adding runways at Heathrow, Gatwick, and Stansted increased capacity by 62%, and operational changes at Heathrow increased capacity by 5%. Tamas (2013) [9] investigated the effects on ROT of aircraft size, the type of runway exit, and IMC/VMC weather conditions by calculating the time spent by each arrival flight. The author used runway capacity model [10] run by using common final approach path length, final approach speeds, average buffer, and ROT. The ROT and IAT at Boston Logan (BOS), New York La Guardia (LGA), Newark (EWR), and Philadelphia (PHL) were statistically estimated for each wake turbulence category (WTC) pair as a case study. The author concluded that the runway capacity should be increased by about 5 to 8% at any of the airports surveyed in the author's study when RECAT was applied. Tamas (2017) [11] also evaluated the impact of RECAT on runway capacity at the system level, focusing on the complex systems of airspace and airport infrastructure. The author proposed the simulation modeling framework taking into account several factors of airfield, aircraft, air traffic control, and weather by updating the author's older model designed with legacy separation rules in mind. The author's study showed that up to 4.9% additional arrival runway capacity can be achieved by a five-category wake turbulence separation system tailored to the typical US fleet mix compared to FAA's traditional category boundaries. For minimizing ROT, which limits the maximum runway throughput, Skorupski et al. (2017) [12] developed a landing roll simulator and presented an analysis showing that the braking methods during landing rolls have an essential impact on runway throughput and airport capacity. They have shown that ROT can be reduced by as much as 50% by using the proposed braking profile. Mascio et al. (2020) [13] created a user-friendly and convenient free spreadsheet-based analytical tool that directly estimates airport runway capacity using ROT and WTC. In their case study, the runway capacity was estimated to be around 8 to 18% higher than the results obtained with FTS[14], as their model did not fully capture the characteristics of the constraints imposed by the airport infrastructure. In order to identify the precursors of an increase in ROT or a runway exit miss, Herrema et al. (2019) [15] proposed a machine learning approach for predicting the runway exit to be used based on actual movements at the airport. The results showed that their proposed model achieved 79% accuracy rate of the decision variable which determined whether a flight rolled out runway exit following the procedure or not. Ahmed et al. (2016) [16] proposed an evolutionary optimization approach to maximize runway capacity for arrival and departure on a single runway. The results showed that their proposed model increased the throughput by 3 per hour and decreased processing time by 200 s compared to First-Come First-Serve (FCFS) conventional approach. Liang et al. (2018) [17] proposed an optimal trajectory planning system with a new route network system for parallel runway arrivals and departures, and the results showed that the proposed system could increase throughput by approximately 26% compared to the baseline. The above studies have contributed to efficient use of the increasing runway capacity. However, the impacts thereof on the traffic flow of aircraft arriving at the destination airport have not been discussed yet. Further, it would be of interest to analyze the impact of the lower IAT on the runway, on the airspace surrounding the destination airport, and even on the arrival

traffic flow as a whole, given the available ROT and airspace capacity in the terminal area, while ensuring reasonable delays in arrival times.

To clarify this, past studies by one of the authors [18–21] presented an analysis of the impact of delay time assuming increasing arrival rate in the terminal and en-route airspace. These studies modeled the arrival traffic flow by means of two types of queuing models, and theoretically analyzed the optimal balance between increasing the arrival rate and considering the airspace capacity, with the ultimate aim of minimizing the delay in arrival time. These queue-based modeling approaches clarified that the application of RECAT would enable us to reduce the arrival delay time while increasing the arrival rate by up to 20%. As the next step, for validating and even enhancing the theoretical results, this study conducts simulation experiments assuming actual operational constraints at the target runway and in the surrounding airspace, and evaluates the impact of reducing IAT at Tokyo International Airport on the arrival traffic flow. A stochastic distribution of ROT at the target runway, which limits the maximum throughput of the runway, is determined based on actual radar track data from the airport.

This paper is organized as follows: Section 2 explains three factors, namely, wake turbulence categories, safety margins, and ROT, which are used in the paper to estimate the maximum arrival capacity of a single runway. Section 3 describes the air traffic arrival operations at Tokyo International Airport. Section 4 describes data-driven estimation of the maximum runway capacity conducted considering the three relevant factors introduced in Section 2. Section 6 proposes simulation models, including airspace and route configurations, operational and flight rules, weather conditions, and runway configurations under the estimated available runway capacity. In addition, data-driven approaches that utilize actual flight plans and radar data from 2019 are employed to estimate the simulation models. A series of simulation experiments were conducted to analyze the impact of reducing the IAT within the arrival traffic flow at the runway and on the point-merge routes in the terminal area. Section 7 discusses the impact of the lower IAT on the traffic flow of aircraft arriving at the airport based on the simulation results. Furthermore, theoretical results in the authors' past studies [18–21] are validated by comparison with the simulation results. Finally, Section 8 presents conclusions and outlines our plans for future work.

## 2. Three Factors Relevant to Estimating the Maximum Runway Capacity

### 2.1. Wake Turbulence Categories

When an aircraft flies through the air, a pair of vortices with different orientations are generated to the rear from the left and right wing tips. These are called wake vortices. Wake vortices generally attenuate over time, move downwards, and are also swept away by the surrounding wind. Since wing-tip vortices are stable and remain in the air for up to three minutes after an aircraft passes, wake vortices are especially dangerous in takeoff and landing situations. Moreover, wake vortices' characteristics depend on various conditions, such as the weight of the aircraft, the wing span, the load distribution, and the weather. To avoid the catastrophic consequences, wake turbulence categories determine aircraft separation minima between the preceding and succeeding aircraft according to their aircraft type.

Several wake turbulence categories [1,2,22] have been suggested depending on the statistics regarding aircraft types at the case study airports, as summarized in one of the authors' past studies [18]. In this paper, the new ICAO standard in [22] is called RE-CAT. Table 1 compares wake turbulence category based on ICAO standard and RECAT. Table 2 summarizes distance-based wake turbulence separation minima (Min. Sep.) between preceding (Pre.) and succeeding (Suc.) aircraft based on wake turbulence categories with the two standards. In Japan, the ICAO standard [23] is currently applied (as of March 2020), and RECAT will replace it in the near future. In RECAT, the currently used ICAO standard is subdivided into six or seven categories according to the maximum takeoff mass (MTOM) and wing span (WS) [22]. Compared with the ICAO standard, RECAT reduces

the separation minima for most combinations of preceding and succeeding aircraft types, as shown in Table 2.

**Table 1.** Comparison of wake turbulence category.

| ICAO Standard | RECAT |
|---|---|
| **SUPER (J)**<br>A380 | **A**<br>136,000 kg $\leq$ MTOM<br>74.68 m < WS |
| **HEAVY (H)**<br>136,000 kg $\leq$ MTOM | **B**<br>136,000 kg $\leq$ MTOM<br>53.34 m < WS $\leq$ 74.68 m<br>**C**<br>136,000 kg $\leq$ MTOM<br>38.10 m < WS $\leq$ 53.34 m |
| **MEDIUM (M)**<br>7000 kg < MTOM < 136,000 kg | **D**<br>18,600 kg < MTOM < 136,000 kg<br>32 m < WS<br>**E**<br>18,600 kg < MTOM < 136,000 kg<br>27.43 m < WS $\leq$ 32 m<br>**F**<br>18,600 kg < MTOM < 136,000 kg<br>WC $\leq$ 27.43 m |
| **LIGHT (L)**<br>MTOM $\leq$ 7000 kg | **G**<br>MTOM $\leq$ 18,600 kg |

**Table 2.** Distance-based wake turbulence separation minima (Min. Sep.) between preceding (Pre.) and succeeding (Suc.) aircraft based on wake turbulence categories with two standards. The blank separation minima and the separation minima of other pairs of aircraft are determined according to Minimum Radar Separation (MRS).

| ICAO Standard | | | RECAT | | |
|---|---|---|---|---|---|
| **Pre.** | **Suc.** | **Min. Sep. (NM)** | **Pre.** | **Suc.** | **Min. Sep. (NM)** |
| J | J | | A | A | 3 |
| | H | 6 | | B | 4 (−2) |
| | | | | C | 5 (−1) |
| | | | | D | 5 (−2) |
| | M | 7 | | E | 6 (−1) |
| | | | | F | 6 (−1) |
| | L | 8 | | G | 8 |
| H | H | 4 | B | B | 3 (−1) |
| | | | | C | 4 |
| | M | 5 | | D | 4 (−1) |
| | | | | E | 5 |
| | | | | F | 5 |
| | L | 6 | | G | 7 (+1) |

**Table 2.** *Cont.*

| ICAO Standard | | | RECAT | | |
|---|---|---|---|---|---|
| **Pre.** | **Suc.** | **Min. Sep. (NM)** | **Pre.** | **Suc.** | **Min. Sep. (NM)** |
| H | M | 5 | C | D | 3 (−2) |
| | | | | E | 3.5 (−1.5) |
| | | | | F | 3.5 (−1.5) |
| | L | 6 | | G | 6 |
| M | L | 5 | D | G | 4 (−1) |

*2.2. Safety Margins Depending on the Levels of Automation Support*

Figure 1 shows the relationship between separation minima and safety margin. The safety margin is defined based on distance or time, and is given in addition to aircraft separation minima for the purpose of ensuring operational safety. The FAA introduced Inter-Aircraft Time Sigma (IAT Sigma) as one of the benchmarks for safety margins [4]. Figure 2 illustrates the IAT Sigma for a aircraft. IAT Sigma corresponds to one sigma (standard deviation (STD)) value in the deviation of aircraft arrival time. The value of IAT Sigma depends on the level of the automation systems which support ATCOs, as provided in Table 3. The four levels of functionality in Table 3 are explained as follows:

1. "No metering" is the baseline capability (i.e., traffic volume is low, and ATCOs are able to handle the operation without any automation support).
2. "Metering" is the capability when ATCOs are supported by Arrival Management (AMAN) systems [24], which automatically provide arrival sequencing and spacing instructions to ATCOs.
3. "GIM-S, TSAS" is the capability when there is support from advanced levels of AMAN [25], which enable more accurate spacing by providing several advisories and visual aids to ATCOs.
4. "IM" is combination of GIM-S and a Flight-deck Interval Management (FIM) system [26], which realizes airborne self-separation of aircraft through the use of speed adjustment.

In general, a higher level of automation support gives a lower value of IAT Sigma. Therefore, implementing higher automation support accommodates increasing arrival rate by reducing IAT.

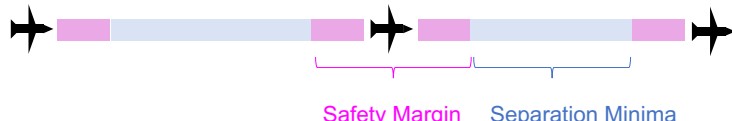

**Figure 1.** Safety margin and separation minima.

**Table 3.** Inter-Aircraft Time Sigma (IAT Sigma) when applying different levels of automation [4].

| Automation Level | IAT Sigma (sec) |
|---|---|
| No metering (baseline) | 18.0 |
| Metering | 16.5 |
| GIM-S, TSAS | 12.0 |
| IM | 5.0 |

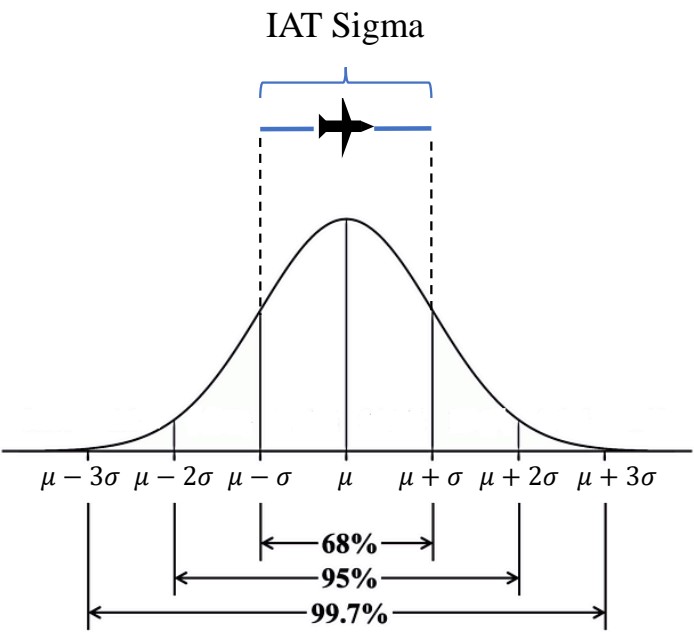

**Figure 2.** Inter-Aircraft Time Sigma (IAT Sigma).

*2.3. Runway Occupancy Time*

Runway Occupancy Time (ROT) is one of the main elements limiting the runway throughput at the airport. Figure 3 demonstrates the section where ROT is calculated. ROT is formally defined as the time interval between when the aircraft crosses the threshold and its tail vacates the runway [27]. Under the regulations in [28], simultaneous occupancy of the runway by multiple landing aircraft is prohibited, and an aircraft cannot be cleared to land if the previous aircraft has not vacated the runway. In Japan, the probability that the succeeding aircraft lands while the preceding aircraft is still on the runway is required to be lower than 0.5% of the total arrivals. This constraint determines the upper bound of the runway throughput.

A past study by one of the authors [18] estimated that the runway throughput of a single runway at Tokyo International Airport (RJTT) could be increased by up to 133% by reducing IAT by applying RECAT and safety margins depending on the level of automation support. However, the maximum limitation (upper bound) caused by the ROT constraints have not been taken into account. Therefore, this paper estimates the maximum throughput of the target runway at the airport by considering ROT as well as RECAT.

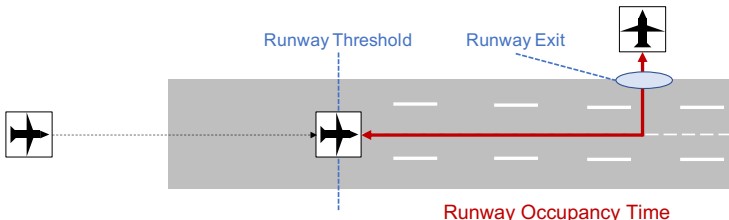

**Figure 3.** Runway Occupancy Time (ROT).

## 3. Description of Case Study Data—Tokyo International Airport

*3.1. Runway Configurations*

RJTT is the busiest airport in Japan, and the fourth busiest airport in the world by passenger traffic in 2017 [29]. For even more efficient operation, implementation of RECAT and a higher-level automation support system at the airport are under discussion.

The airport uses four runways: a set of parallel north-south runways (34L/16R and 34R/16L) and two southwest-northeast crosswind runways (22/04 and 23/05). Figure 4

shows the runway configuration for departures and arrivals at RJTT. Two major runway configurations are used depending on the wind direction. Figure 4a depicts northerly wind operation. In northerly wind operation, aircraft arrive at either runway 34L or runway 34R, and departing aircraft take off from runway 05 or runway 34R, depending on their origin/destination airports . Basically, northbound traffic uses runway 34R for both departure and arrival, while southbound traffic uses runway 05 for departure and runway 34L for arrival. On the other hand, Figure 4a depicts northerly wind operation. Southerly wind operation, in which aircraft arrive at runway 16L and 16R , started in 2020. In this operation, departure aircraft bound for south-western destinations basically take off from runway 22, and the others depart from runway 16L and runway 16R in combination with arrival aircraft.

Climate and seasonal fluctuations cause changes to runway configurations. According to the records for operations between 6:00 a.m. and 11:00 p.m. for 2016 to 2018, the northerly wind operation accounted for 70% of the total, taking a larger share than the southerly wind operation. Based on these statistics, this paper focuses on the northerly wind operation.

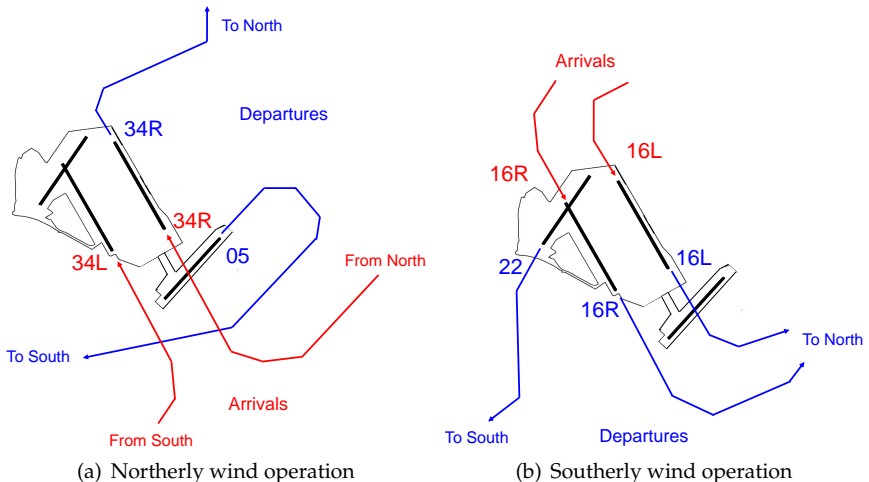

    (a) Northerly wind operation        (b) Southerly wind operation

**Figure 4.** Runway configuration for departures (**blue lines**) and arrivals (**red lines**) at RJTT.

### 3.2. Arrival Operations

Figure 5 shows one day's tracks of arrival air traffic at RJTT under the northerly wind operation with the nominal traffic conditions. In this paper, a series of simulation experiments is conducted using actual radar track and flight plan (FP) data. Arrival flights were extracted from FPs and track data for a period of 39 days selected from one week of each month between September 2019 and February 2020. All the data cover the nominal operation at RJTT that excludes impacts from weather and other rare events. There were, on average, 623 arrivals per day: 502 domestic and 121 international. With respect to the arrival runway, 472 flights coming from a south-western direction landed on runway 34L, and 151 flights coming from a northern direction landed on runway 34R. This means that the number of arrivals at runway 34L is more than three times that of arrivals at runway 34R.

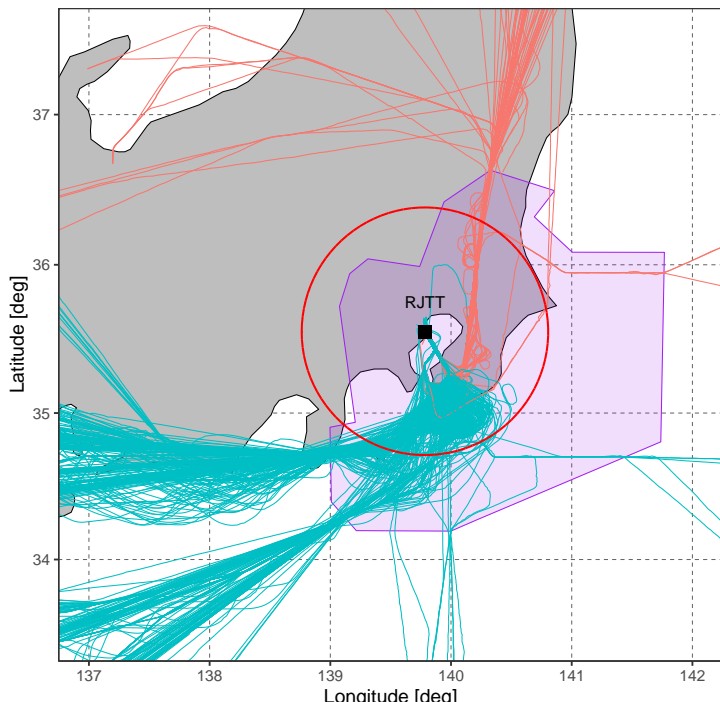

**Figure 5.** Actual radar tracks bound for RJTT coming from a northern direction (orange lines) and southern direction (blue lines) in the northerly wind operation in a day in October 2019. Tokyo Approach Control Area (TACA) is bounded by a purple polygon. The radius of the red circle is 50 NM centered on RJTT.

In the arrival traffic at runway 34L and 34R, one specific feature we focused on is point-merge (PM) operation in the terminal airspace. PM operation at RJTT started in July 2019. Figure 6 shows the configuration of PM routes applied in Tokyo Approach Control Area (TACA) under northerly wind operation. ML-PM1 is composed of three overlapping PM routes located in the southern area of TACA, whereas PM2 is composed of a single PM located in the northern area of TACA. As mentioned in Section 3.1, RJTT applies an operation in which inbound traffic coming from southern and northern directions land on runway 34L and 34R, respectively. Therefore, ML-PM1 and PM2 are basically segregated and independent. With respect to ML-PM1, three arrival flows come into the vertically and laterally separated arcs (Sequencing Legs), finally merging at the Merge Point. The Sequencing Legs are dedicated arcs for path stretching/shortening using Turning Points at which the aircraft turn to the Merge Point. The operating method is composed of two steps [30]:

- Spacing arrivals by "direct-to" instructions from the Turning Points to the Merge Point. The spacing distance or time is controlled by selecting Turning Points on the Sequencing Legs: an arrival aircraft flies a longer distance on the Sequencing Legs if a larger spacing is required between it and the preceding arrival aircraft.
- Maintaining the spacing by speed control after leaving Sequencing Legs.

Descent instructions are given by ATCOs when the arrivals leave the Sequencing Legs.

Application of the PM operation is expected to reduce the costs of communication between pilots and ATCOs. Arrival delay is controlled via holding operation before entering the PM routes when the required arrival spacing is larger than the PM operation is capable of.

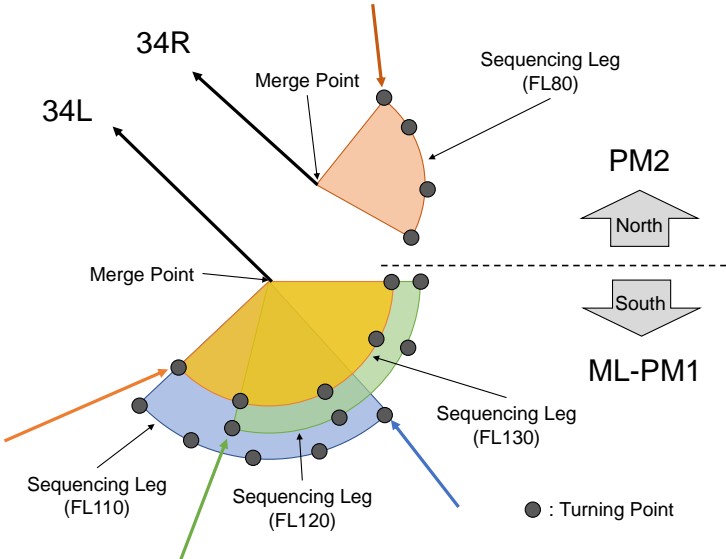

**Figure 6.** Schematic diagram of the configuration of the point-merge (PM) system connected to 34L (multi-level point merge 1: ML-PM1) and 34R (point merge 2: PM2) in Tokyo Approach Control Area (TACA) with northerly wind operation.

## 4. Estimating Available Runway Capacity

### 4.1. Estimating the Maximum Arrival Rate

This chapter estimates the maximum arrival rate (number of arrival aircraft per hour) at runway 34L, $\lambda_{RW34L}$, based on two wake turbulence categories shown in Tables 1 and 2 and safety margins depending on different levels of automation categorized in Table 3 . We determine this arrival rate as follows:

$$\lambda_{RW34L} = \frac{3600.0}{\mathbb{E}[t_{IAT}]}. \tag{1}$$

Here, $\mathbb{E}[t_{IAT}] = \mathbb{E}[t_{min}] + t_{\sigma}$, where $\mathbb{E}[t_{min}]$ is the mean of the minimum time separation in seconds corresponding to each wake turbulence category. In this paper, $t_{min}$ corresponds to the minimum time separation at runway 34L in the northerly wind operation during the congested time period 5:00–10:00 p.m. The minimum distance separation in Table 2 was divided by the ground speed of each arrival aircraft at the runway threshold. The ground speed was estimated using actual radar data on the surface traffic we will explain the data in Section 4.2. The stochastic distribution of $t_{min}$ when RECAT is applied is shown in Figure 7. $t_{\sigma}$ is IAT Sigma in seconds corresponding to each automation level (see Table 3).

Table 4 summarizes the estimated $\mathbb{E}[t_{IAT}]$ and $\lambda_{RW34L}$ when RECAT is applied under the four levels of automation support. Currently, runway 34L allows approximately 30 arrivals an hour (every 2 min). Compared with the current operation, for example under 'No metering' (with no automation support), 5 more aircraft (a total of 35 arrivals an hour) will be able to arrive at runway 34L if RECAT is applied to the operation at RJTT. Higher levels of automation support give even larger $\lambda_{RW34L}$ by reducing the safety margins added to the minimum separations. The highest level of automation (IM) is expected to give 11 more arrivals (a total of 41 arrivals) per hour.

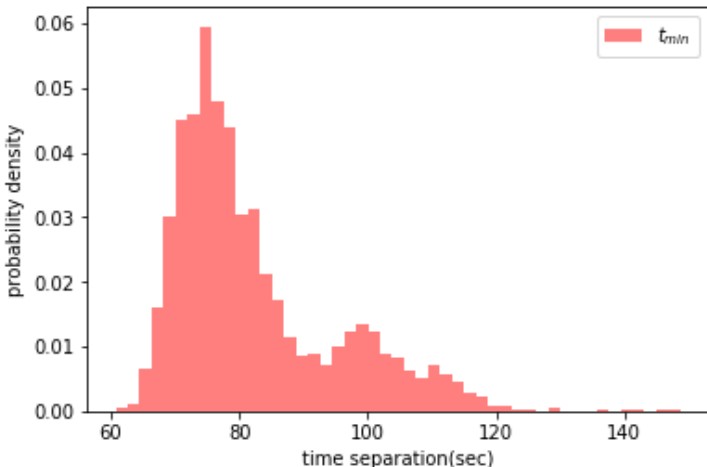

**Figure 7.** Estimated $t_{min}$ distribution at runway 34L during the peak period.

**Table 4.** The maximum arrival rate $\lambda_{RW34L}$ under RECAT separation depending on the level of automation support.

| Automation Level | $\mathbb{E}[t_{IAT}]$ | $\lambda_{RW34L}$ (ac/hour) |
|:---:|:---:|:---:|
| No metering | 100.3 | 35 |
| Metering | 98.81 | 36 |
| GIM-S,TSAS | 94.31 | 38 |
| IM | 87.31 | 41 |

*4.2. ROT Constraints*

As discussed in the previous chapter, RECAT application will be effective for increasing the maximum arrival rate at runway 34L under automation support. However, as shown in Equation (1), the maximum limitation of runway throughput has not been considered yet. For operational safety, it is necessary to limit the probability of an aircraft landing at the runway while the preceding aircraft is still on it. In other words, IAT at the runway threshold is required to be larger than the ROT of the preceding aircraft to avoid simultaneous runway occupancy with a high probability.

Considering the ROT constraints, maximizing the mean runway throughput per hour $\mathbb{E}[\tau]$ is stated using the probability $P$ as follows:

$$\text{maximize} \qquad \mathbb{E}[\tau]$$
$$\text{subject to} \qquad P\{t_{IAT} < t_{ROT}\} \leq \alpha_R.$$

$t_{ROT}$ is the ROT of the preceding aircraft in seconds during the corresponding time period. Here, $t_{IAT} = t_{min} + t_\sigma$. $\alpha_R$ is the acceptable probability of missed approach, and 0.5% is given in Japan as mentioned in Section 2.3.

Figure 8 shows example probability density distributions of $t_{ROT}$ (in blue) and $t_{IAT}$ (in red) at runway 34L. The distributions are given by actual radar tracks of the aircraft surface movements in the peak period (5:00–10:00 PM) for 95 days between May 2016 and March 2018 at RJTT (the data sets for 2018 were the newest ones). The IAT distribution corresponds to $t_{IAT}$ when the RECAT separation minima are applied to $t_{min}$ and $t_\sigma = 0$. The overlap in the ROT and IAT distribution is required to be less than $\alpha_R = 0.005$ [31].

In line with this, the maximum runway throughput $\mathbb{E}_{max}[\tau]$ of arrivals at runway 34L is defined as follows:

$$\mathbb{E}_{max}[\tau] \equiv \sup\{\hat{\lambda}_{RW34L} | P\{t_{IAT} < t_{ROT}\} \leq \alpha_R\}. \tag{2}$$

Here, $\hat{\lambda}_{RW34L}$ is the estimated arrival rate below. To calculate $\mathbb{E}_{max}[\tau]$, we use the following process:

1.  Extract the IAT data for arrival aircraft pairs for which the separation minima are 3 NM, which is the shortest separation minimum at RJTT, and make a data set of $t_{IAT}$. The 3 NM pairs account for more than 70% of all the arrivals (see Table 5) and have the potential to violate the ROT constraints.
2.  Identify the stochastic distribution of ROT and IAT (corresponding to the 3 NM separation given by the previous step) using a normal distribution and a gamma distribution, respectively [31].
3.  Calculate the percentage of the overlap in the ROT and IAT distributions. Since this only represents the proportion in relation to the aircraft pairs with 3 NM separation, convert it to the proportion in relation to all the aircraft pairs using the data in Table 5.
4.  Adjust the mean of the IAT distribution, $\mathbb{E}[t_{IAT}]$, until the maximum value of $\hat{\lambda}_{RW34L}$ that satisfies having an overlap of less than 0.5% is found using Equation (1). ($\mathbb{E}[t_{IAT}]$ is the denominator in Equation (1).)

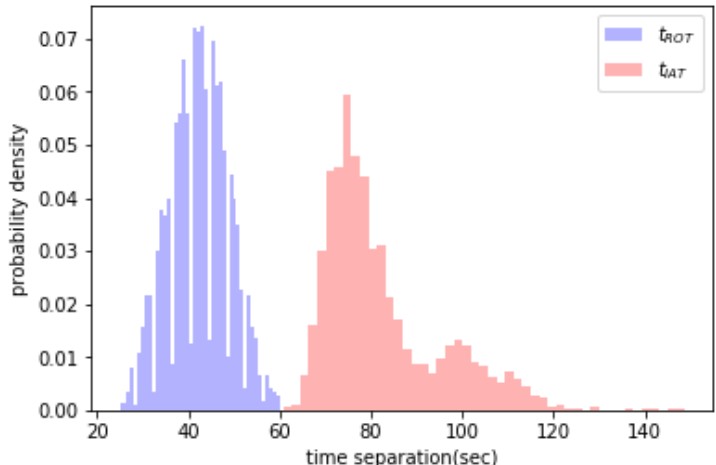

**Figure 8.** Probability density distributions of $t_{ROT}$ and $t_{IAT}$.

**Table 5.** Statistics for RECAT separation minima applied for aircraft pairs arriving at RJTT.

|                 | 3 NM | 3.5 NM | 4 NM | 5 NM+ |
|-----------------|------|--------|------|-------|
| Total quantity  | 2717 | 8      | 774  | 14    |
| Rate (%)        | 77.4 | 0.2    | 22.0 | 0.4   |

We repeated this procedure for the combinations of two wake turbulence categories and ROT constraints. In addition, the mean and standard deviation (std) of the ROT distributions were adjusted in the steps to study the impact of the ROT constraints on the maximum runway throughput, $\mathbb{E}_{max}[\tau]$. Figure 9 shows the 9 cases of $t_{ROT}$ distributions and $t_{IAT}$ distributions of arrival aircraft pairs with a separation minimum of 3 NM that maximize the runway capacity while satisfying the ROT constraints in each case.

Table 6 shows the estimation results for 9 combinations of ROT mean and std. When all $t_{ROT}$ values follow the distribution in the actual radar data on the surface operation (the blue distribution in Figure 8), $\mathbb{E}[t_{IAT}]$ is 87.28 s, allowing a maximum of 41 arrival aircraft an hour. When the mean and std of ROT are larger, for instance because of bad weather conditions, $\mathbb{E}_{max}[\tau]$ is reduced. For instance, if the mean of ROT increases by 25% of that of the original data, $\mathbb{E}[t_{IAT}]$ increases to 95.51 s and $\mathbb{E}_{max}[\tau]$ is limited to 37 aircraft an hour.

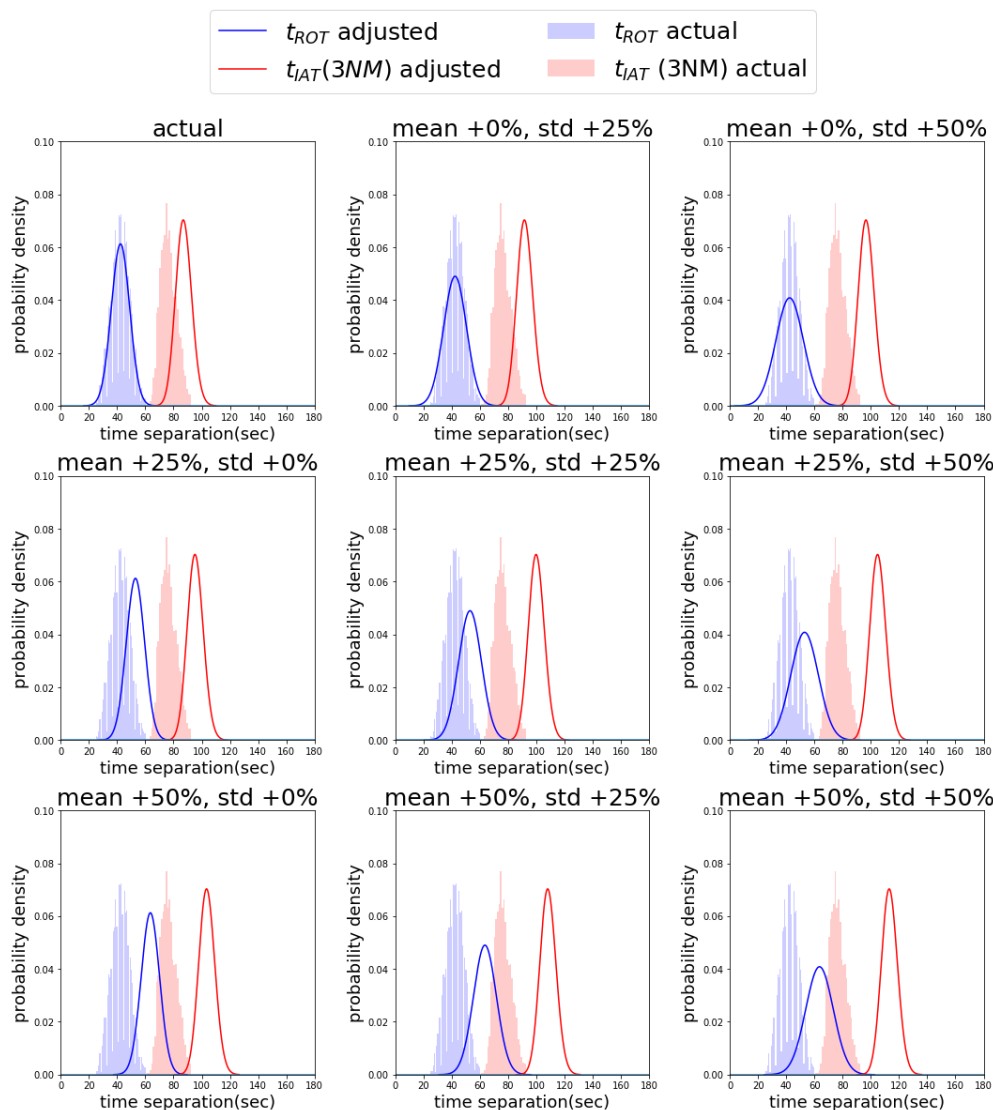

**Figure 9.** Nine cases of $t_{ROT}$ distributions (**blue**) and $t_{IAT}$ distributions of arrival aircraft pairs with a separation minimum of 3 NM (**red**) that maximize the runway capacity while satisfying the ROT constraints in each case.

**Table 6.** Estimating the maximum runway capacity according to the ROT distribution.

| ROT Mean | ROT Std | $\mathbb{E}[t_{IAT}]$ (sec) | $\mathbb{E}_{max}[\tau]$ (ac/hour) |
|---|---|---|---|
| +0% | +0% | 87.28 | 41 |
| | +25% | 91.93 | 39 |
| | +50% | 96.84 | 37 |
| +25% | +0% | 95.51 | 37 |
| | +25% | 100.16 | 35 |
| | +50% | 105.05 | 34 |
| +50% | +0% | 103.73 | 34 |
| | +25% | 108.38 | 33 |
| | +50% | 113.28 | 31 |

*4.3. Available Runway Capacity*

Here, we define the available runway capacity at runway 34L, $\mathbb{E}_{RW34L}[\tau]$, using $\lambda_{RW34L}$ and $\mathbb{E}_{max}[\tau]$ given in Equations (1) and (2), as follows:

$$\mathbb{E}_{RW34L}[\tau] \equiv \min\{\lambda_{RW34L}, \mathbb{E}_{\max}[\tau]\}. \tag{3}$$

Figure 10 compares $\lambda_{RW34L}$, when RECAT (green line) and ICAO standards (red line) are applied under the four levels of automation support, and $\mathbb{E}_{\max}[\tau]$ (black lines) estimated using the procedure in Section 4.2 according to the ROT constraints summarized in Table 6. The most significant point is that the maximum arrival rate, $\lambda_{RW34L}$, achieved by RECAT application under IM (the highest level of automation support) is 41 arrivals an hour, which is the same value as $\mathbb{E}_{\max}[\tau]$ . That value is estimated using ROT data in the actual radar tracks, corresponding to the value with which ROT mean and ROT std are +0% in Table 6.

This means that any combinations of the RECAT application and proposed automation support will be fully effective for increasing the runway capacity under the current aircraft types and arrival sequencing at RJTT, without breaking the ROT constraints.

Table 7 summarizes the available runway capacity at runway 34L per hour, $\mathbb{E}_{RW34L}[\tau]$, corresponding to the RECAT and ICAO standards. The current RJTT operation applies "Metering" automation support under the ICAO standards, so 4 more arrivals will be allowed by applying RECAT even if the same levels of automation support are kept (RECAT application will allow 36 arrivals an hour, and the ICAO standards allow 32 arrivals an hour). Based on these results, the next chapter runs a series of simulation experiments and evaluates the impacts on the arrival traffic flow when 4 more aircraft arrive at RJTT an hour. To evaluate the impacts of increasing arrivals, the arrival traffic flow is modeled in the next chapter.

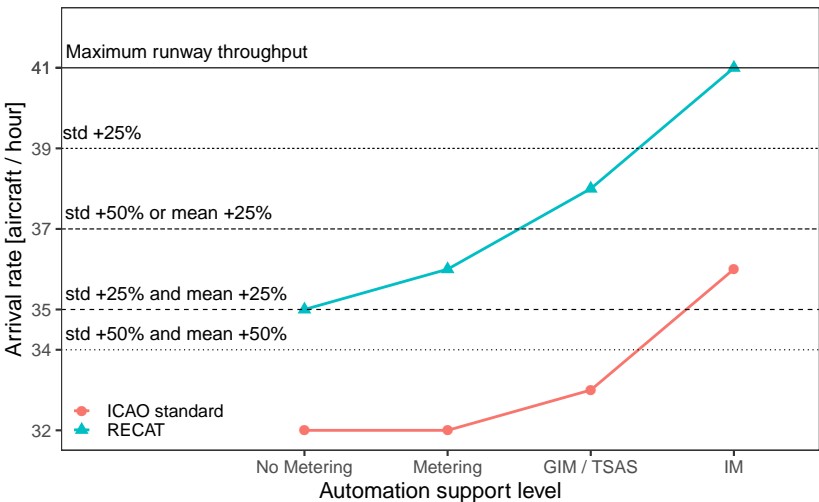

**Figure 10.** Estimation of the available runway capacity.

**Table 7.** Available runway capacity under the ROT constraints.

| Automation Level | $\mathbb{E}_{RW34L}[\tau]$ (ac/hour) | |
| :---: | :---: | :---: |
| | **RECAT** | **ICAO Standards** |
| No metering | 35 | 32 |
| Metering | 36 | 32 |
| GIT-S, TSAS | 38 | 33 |
| IM | 41 | 36 |

## 5. Modeling Arrival Traffic Flow

### 5.1. Traffic Scenarios

Traffic scenarios for arrivals at RJTT are modeled based on data sets in the actual FPs and radar tracks, corresponding to arrival operation as updated on July 2019. The data are summarized in Section 3.2. The traffic scenarios include the following information: call sign, reference time, cruising altitude, aircraft type, and flight path.

Reference time means the scheduled time of departure at the airport, or the crossing time at the initial fix in the simulation. The departure time of pop-up flights significantly impacts the arrival traffic flow. This is especially true of those departing from Osaka International Airport (RJOO), Kansai International Airport (RJBB) and Chubu International Airport (RJGG), located 150 to 200 NM away from RJTT Figure 11 shows the distance from RJTT and Standard Instrumental Departures (SIDs) for RJOO, RJBB and RJGG. Major impacts are caused by pop-up flights departing from RJOO, a total of 30 aircraft on average per day. Spacing adjustments could often occur when the departures from RJOO merge into the routes on which it is congested mainly due to the flights from RJFF, which hold the largest number of RJTT arrivals.

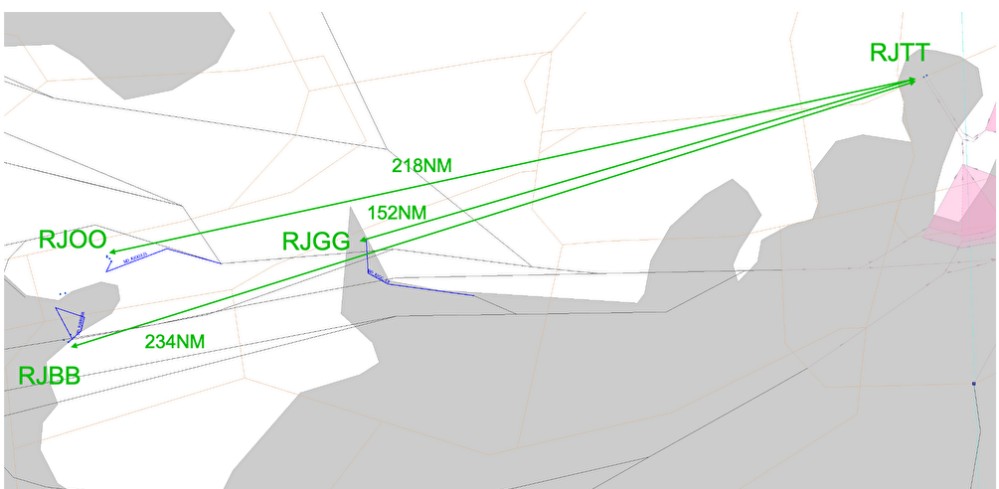

**Figure 11.** Distance from Tokyo International Airport (RJTT) (**green lines**) and Standard Instrumental Departures (SIDs) (**blue lines**) for Osaka International Airport (RJOO), Kansai International Airport (RJBB) and Chubu International Airport (RJGG).

The maximum altitude recorded in the FPs was employed to define the cruising altitude of each arrival aircraft. Statistics on aircraft types are realized in the simulation according to the actual radar data, because they decide IAT as discussed in Section 2. Figure 12 demonstrates the flight paths of arrival traffic at RJTT in a day. As shown in the figure, actual flight paths in the en-route airspace of Fukuoka FIR were modeled in the simulation scenarios.

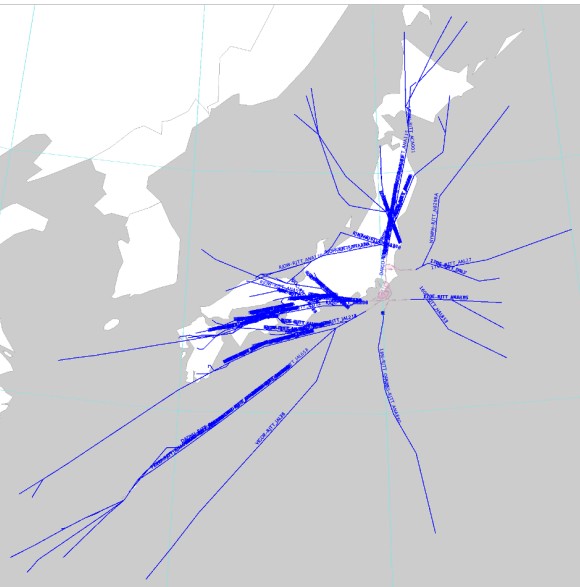

**Figure 12.** Flight paths of arrivals at RJTT in a day (**blue lines**), excluding standard terminal arrival routes (STARs) and standard instrumental departures (SIDs).

### 5.2. Airspace and Routes

This simulation targets all arrival traffic at RJTT in Fukuoka FIR, which is under the control of the Japanese Civil Aviation Bureau (JCAB). Thus, coordinate data for the sectors, fixes and runways in the FIR are obtained from the Aeronautical Information Publication (AIP) published by JCAB [32].

All en-route sectors handled by area control centers and Tokyo Approach Control Area (TACA) are modeled in the simulation. Figure 13 shows the TACA boundary as a polygon. In TACA, standard terminal arrival routes (STARs) in daytime operation are modeled with speed and altitude constraints at designated fixes. As shown in Figure 13, point-merge routes at runway 34L and 34R (see Figure 6) are assigned for the STARs. A total of 6 fixes are given as transition points, and in-trail separation at each point is assigned to arrivals in terms of distance.

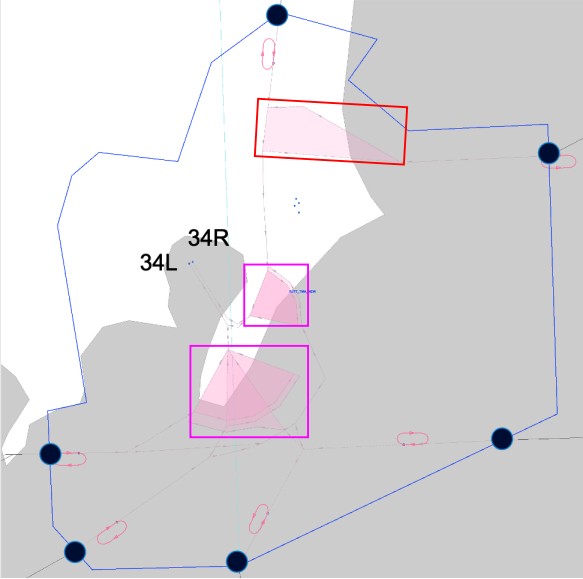

**Figure 13.** TACA (blue polygon) and two runways (34L and 34R) at RJTT. Vectoring zones are composed of an red rectangle (radar vector) and pink rectangle (point-merge system). The traffic volumes in TACA and in-trail separation at entry fixes are assumed in the simulation.

### 5.3. Separation Rules

Figure Separation rules are applied for arrivals. To realize the arrival traffic volume in TACA and the in-trail separation distance at 6 transition points, we model in-flight and departure time control in the operation. Holding zones are located close to the fixes and ensure the assigned in-trail distance at each fix as shown in Figure 13 .

Two types of wake turbulence categories summarized in Tables 1 and 2 are given as aircraft separation minima. 5 NM and 3 NM radar separations are ensured in en-route sectors and TACA, respectively.

### 5.4. Arrival Traffic Volume

Based on the discussion in Section 4.3, two arrival rates at runway 34L and 34R are employed as arrival traffic volumes in TACA. In the arrival operation as updated in July 2019, 28 arrivals[/hr] and 12 arrivals[/hr] are allowed to land on runway 34L and 34R, respectively. Compared to the previously applied arrival operation, these values are conservative settings for safety reasons, especially taking into consideration runway crossing at 34L.

According to the updated arrival operation, a total of 40 arrivals are assumed at RJTT an hour. Thus, 40 arrivals are given as the actual volume in TACA. Section 4.3 concludes that having 4 more arrivals at runway 34L is acceptable. According to this, we assume a total of 44 arrivals at RJTT an hour.

### 5.5. Weather Information

All the simulation experiments assume nominal weather conditions with no specific impacts on the air traffic operation. For wind information, we employ the Meso Scale Model Grid Point Value (MSMGPV) provided by the Japan Meteorological Agency [33]. The data includes atmospheric properties such as wind and temperature on a three-dimensional grid, and are published every 3 hours. The grid points are located every 0.125 degrees of longitude and 0.1 degrees of latitude at every 50–100 hPa pressure altitude. In the simulation, the wind speed and direction at the grid point closest to each aircraft is used to calculate the ground speed.

## 6. Evaluating the Impacts of Increasing Arrivals

### 6.1. Simulation Environment

We implemented the model designed in Section 5 in the AirTOp software [34], which is a fast-time simulation platform based on multi-agent models that enable gate-to-gate simulation of air traffic. Each agent moves following the BADA model [35] according to the status of each aircraft, such as on-ground, climb, cruise, and descent. AirTOp is one of the standard simulation platforms used by air navigation service providers (ANSPs) and research institutes in the fields of air traffic management (ATM) for discussing relevant operational challenges in the future. Although implementing advanced models in AirTOp is challenging, the benefit comes from getting various outputs, including 4D trajectories, en-route and terminal delay time, and fuel consumption.

### 6.2. Arrival Separation at Runway 34L

As shown in Table 8, four simulation scenarios, Case 1 to 4, cover combinations of wake turbulence categories and arrival traffic volumes in TACA the following Sections 5.3 and 5.4.

Case 1 and 2 adopted ICAO standard while Case 3 and 4 utilized RECAT. When ICAO standard is replaced by RECAT, a total of 40.8% of arrival aircraft pairs had reduced IAT at runway 34L, as shown in Table 9; 32.6 % of the total had IAT reduced by 1.0 NM, and 8.1% had it reduced by 2.0 NM. The rate for 1.5 NM reduction is less than 0.1% because small aircraft categorized into class E and F in RECAT summarized in Table 1 rarely arrive at RJTT.

Figure 14 compares IAT distribution at runway 34L for the ICAO standards (Case 1 and 2) and RECAT (Case 3 and 4). It seems that the RECAT results have a longer tail distribution

than the ICAO's because these two tail distributions are overlapped, but the tail lengths are almost the same. This shows that the number of arrival pairs at the peak IAT values, between 110 and 120 s, increased by approximately 40% in total. Even though the arrival rate is the same as the current operation (40 arrivals in total), RECAT contributes to reducing IAT. As shown in Section 4, IAT reduction is advantageous to efficient runway operation.

**Table 8.** Combination of wake turbulence category and arrival traffic volume.

| Simulation Scenario | Wake Turbulence Category | Arrival Traffic Volume in TACA [/hr] |
| :---: | :---: | :---: |
| Case 1 | ICAO standard | 40 |
| Case 2 | ICAO standard | 44 |
| Case 3 | RECAT | 40 |
| Case 4 | RECAT | 44 |

**Table 9.** Rate of aircraft pairs for which separation minima were reduced by applying RECAT at runway 34L.

| Reduced Separation [NM] | Rate [%] |
| :---: | :---: |
| 0.0 | 59.2 |
| 1.0 | 32.6 |
| 1.5 | 0.1 |
| 2.0 | 8.1 |

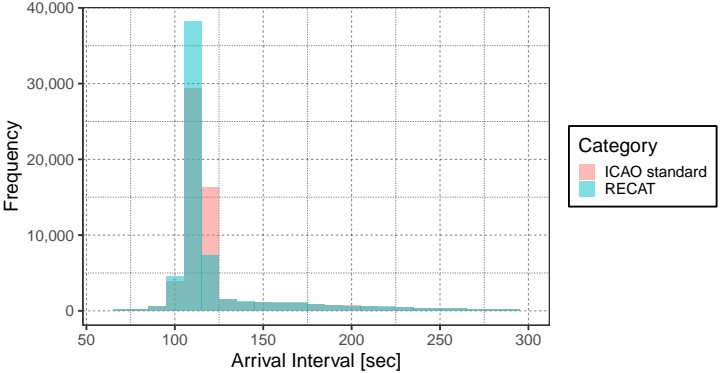

**Figure 14.** Comparing IAT at runway 34L.

*6.3. Vectoring Delay in TACA*

The simulation results clarified that RECAT application will significantly contribute to reducing vectoring time in TACA.

Table 10 and Figure 15 compare arrival delays caused by vectoring on ML-PM1 depicted in Figure 6 . In Table 10, $V_{TACA}$ denotes the arrival traffic volume in TACA. $N_{i,VC}$ is total number of flights vectored on ML-PM1 in Case $i \in \{1, 2, 3, 4\}$. $\mathbb{E}[N_{i,VC}]$ is the mean of $N_{i,VC}$ per day. $\mathbb{E}[D_{i,VC}]$ and $\mathbb{M}[D_{i,VC}]$ are the mean and median vectoring delay per flight, respectively, in Case $i \in \{1, 2, 3, 4\}$.

As shown in Figure 15, comparing Cases 1 and 3, $\mathbb{E}[D_{i,VC}]$ is reduced by 8.19 s if RECAT replaces the ICAO standards for 40 arrivals an hour. For 44 arrivals an hour, comparing Cases 2 and 4, $\mathbb{E}[D_{i,VC}]$ is reduced by 10.30 s.

As shown in Table 10, $N_{i,VC}$ was reduced when RECAT was applied for both 40 and 44 arrivals an hour (Case 3 and 4), compared with application of the ICAO standards with

the same arrival rate (Case 1 and 2). This significantly reduced the total delay time in the arrival traffic flow in TACA.

**Table 10.** Summary of vectoring delay per flight on ML-PM1 depicted in Figure 6.

| *i* (Case Number) | 1 | 2 | 3 | 4 |
|:---:|:---:|:---:|:---:|:---:|
| Wake turbulence category | ICAO standards | | RECAT | |
| $V_{TACA}$ [ac/hr] | 40 | 44 | 40 | 44 |
| $N_{i,VC}$ [ac] | 5810 | 5976 | 5678 | 5846 |
| $\mathbb{E}[N_{i,VC}]$ [ac/day] | 149 | 153 | 146 | 150 |
| $\mathbb{E}[D_{i,VC}]$ [sec/ac] | 164.43 | 179.34 | 156.24 | 169.04 |
| $\mathbb{M}[D_{i,VC}]$ [sec/ac] | 153 | 177 | 142 | 158 |

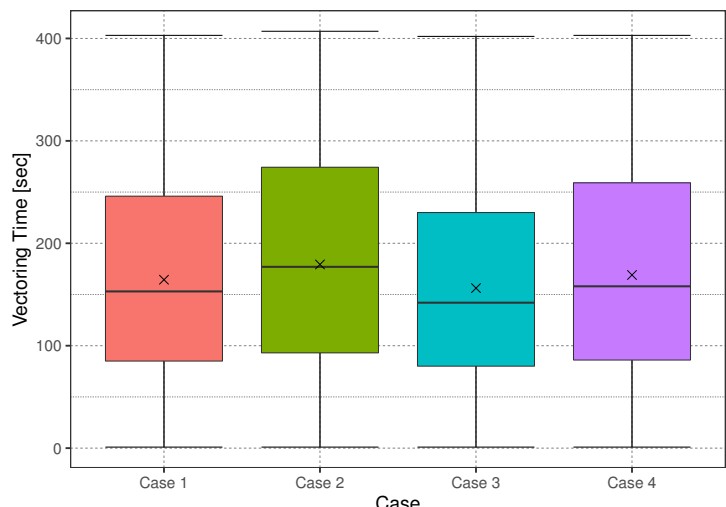

**Figure 15.** Comparison of vectoring delay per aircraft on ML-PM1.

*6.4. En-Route and Takeoff Delay*

The simulation results showed that RECAT application will reduce the number of flights delayed en-route, including holding stack  depicted in Figure 13 and at the departure airport. Table 11 and Figure 16 compare the total arrival delay time en-route and at the departure airport (excluding the vectoring delay).

In Table 11, $N_{i,ER/TO}$ is the total number of flights delayed en-route and in the takeoff phase in Case $i \in \{1, 2, 3, 4\}$. $\mathbb{E}[N_{i,ER/TO}]$ is the mean of $N_{i,ER/TO}$ per day. $\mathbb{E}[D_{i,ER/TO}]$ is the mean en-route and takeoff delay per flight while $\mathbb{M}[D_{i,ER/TO}]$ is the median en-route and takeoff delay per flight in Case $i \in \{1, 2, 3, 4\}$.

More significantly, increasing the arrival rate (capacity in TACA) contributes to reducing the delay time en-route and at the departure airport. Comparing Case 1 and 2 and Case 3 and 4, $\mathbb{E}[D_{i,ER/TO}]$ are reduced by approximately 5 min. These results show that the maximum limitation on runway throughput is the bottleneck in arrival traffic flow, and increasing the capacity at the runway and in the terminal airspace mitigates the delay in the takeoff and en-route airspace.

**Table 11.** Summary of en-route and takeoff delays [min/ac].

| $i$ (Case Number) | 1 | 2 | 3 | 4 |
|---|---|---|---|---|
| Wake turbulence category | ICAO standards | | RECAT | |
| $V_{TACA}$ [ac/hr] | 40 | 44 | 40 | 44 |
| $N_{i,ER/TO}$ [ac] | 14,122 | 13,101 | 14,031 | 12,887 |
| $\mathbb{E}[N_{i,ER/TO}]$ [ac/day] | 362.1 | 335.9 | 359.8 | 330.4 |
| $\mathbb{E}[D_{i,ER/TO}]$ [min/ac] | 8.38 | 3.29 | 8.37 | 3.15 |
| $\mathbb{M}[D_{i,ER/TO}]$ [min/ac] | 4.38 | 2.37 | 4.28 | 2.3 |

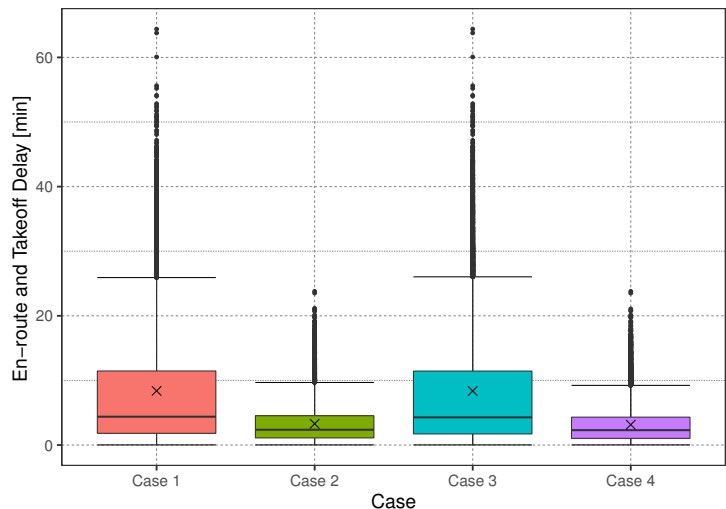

**Figure 16.** Comparison of en-route and takeoff delays.

*6.5. Total Delay in the Arrival Traffic Flow*

As discussed in Sections 6.2 and 6.3, applying RECAT and increasing the airspace and runway capacity reduce the arrival delay time. Here, we compare the total delay in the arrival traffic flow as a whole.

As shown in Table 11 and Figure 16, overall, Case 4 achieved the minimum delay time; this means that IAT reduction is a powerful solution for reducing arrival delay time under conditions of increasing arrival rate and TACA capacity imposed on the available runway capacity. Table 12 and Figure 17 summarize the total arrival delay time in the arrival traffic as a whole as percentages in relation to Case 1 (current operation at RJTT) as the baseline. The percentage, $P_{i,VC}$, for Case $i \in \{1, 2, 3, 4\}$ is the total vectoring delay time obtained by summing over the vectoring delay times of each aircraft, $t_{j,VC}$, $j \in \{1, 2, \cdots, N_{i,VC}\}$, as follows:

$$P_{i,VC} = \frac{\sum_{j=1}^{N_{i,VC}} t_{j,VC}}{\sum_{j=1}^{N_{1,VC}} t_{j,VC}} \times 100.0. \tag{4}$$

Similarly, the percentage for the total en-route and takeoff delay, $P_{i,ER/TO}$, for Case $i \in \{1, 2, 3, 4\}$ is calculated as follows:

$$P_{i,ER/TO} = \frac{\sum_{j=1}^{N_{i,ER/TO}} t_{j,ER/TO}}{\sum_{j=1}^{N_{1,ER/TO}} t_{j,ER/TO}} \times 100.0. \tag{5}$$

In Case 2 and Case 4, with increased runway and TACA capacity, $P_{i,ER/TO}$ are dramatically reduced by more than 60% compared with Case 1.

With the arrival traffic volume in the current operation, RECAT application reduced $P_{i,VC}$ by 7.79% in TACA, and $P_{i,ER/TO}$ by 1.31%, compared with Cases 1 and 3. This shows that RECAT will be effective for reducing total delay time in the arrival traffic as a whole.

Most interestingly, $P_{i,VC}$ in Case 1 is almost identical to that in Case 4 while allowing 10% of additional capacity in TACA and significantly reducing $P_{i,ER/TO}$. This result indicates that applying RECAT and increasing the capacity in TACA under the available runway capacity at RW34L is the most powerful solution for reducing the total delay in the arrival traffic flow as a whole at RJTT.

**Table 12.** Summary of the percentage of the total delay for flights arriving at runway 34L in RJTT compared with Case 1.

| $i$ (Case Number) | 1 | 2 | 3 | 4 |
|---|---|---|---|---|
| Wake turbulence category | ICAO standards | | RECAT | |
| $V_{TACA}$ [ac/hr] | 40 | 44 | 40 | 44 |
| $P_{i,VC}$ [%] | 100 | 109.62 | 92.21 | 100.32 |
| $P_{i,ER/TO}$ [%] | 100 | 37.29 | 98.69 | 34.82 |

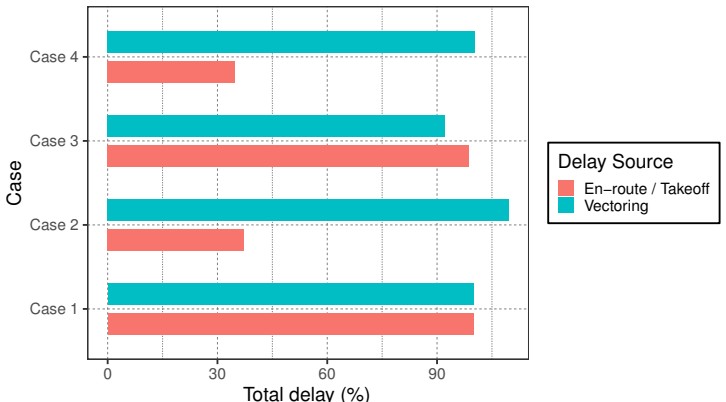

**Figure 17.** Comparing the increase in the total delay in arrival air traffic as a whole at runway 34L in RJTT.

## 7. Discussion

This paper clarified that a lower aircraft separation would significantly reduce total delay time in the arrival traffic as a whole during congested time periods at the target airport. Firstly, the available runway capacity was estimated using IAT and ROT distributions identified in the data from actual traffic arriving at runway 34L of RJTT. Secondly, the arrival traffic flow was modeled based on the data-driven estimation, and implemented in the simulation environment. A series of simulation experiments was conducted and clarified effective arrival strategies under the relevant air traffic operation conditions.

The results showed that the available runway capacity at runway 34L (defined in Equation (3)) allowed an increase to 36 arrivals an hour with RECAT application and the current automation support capability at RJTT, following a First-Come First-Served (FCFS) protocol, which is currently used in air traffic control. In other words, it is possible to achieve the maximum runway throughput without optimizing arrival sequencing in in-flight operation, when the assigned runway capacity is less than 36 aircraft an hour and there is the current level of automation support. Introducing IM automation support satisfying the ROT constraints will make it possible to process a maximum of 41 aircraft an hour.

The simulation results showed that allowing 4 more arrivals an hour at RJTT reduced arrival delay by more than 5 min per aircraft, when IAT was reduced by replacing the

ICAO standards with RECAT. In the terminal area, vectoring time was reduced by approximately 7% to 10% on the PM routes compared with the current operation. These simulation results validated the theoretical results obtained using queuing models in [18–21], which indicated that the lower IAT with the available arrival traffic volume following FCFS sequencing mitigated traffic congestion and reduced arrival delay time compared to the current performance.

## 8. Concluding Remarks

This study modeled arrival traffic flow at an airport based on data-driven analysis. The developed models were implemented in simulation environments, and a series of simulation experiments was conducted to quantitatively clarify the impacts of the lower arrival aircraft separation with the available runway and airspace capacity. The simulation results quantitatively clarified that (1) the IAT reduction significantly reduced terminal delay time and (2) increasing airspace capacity in terminal area drastically reduced the en-route delay time in the arrival traffic flow. These results were equivalent to results based on queuing theory obtained in past studies by the authors, and therefore validated the theoretical results. The authors' main challenge is to propose a scientific and systematic way of designing future air traffic management by integrating data analytic, theoretical modeling and simulation evaluation. This paper presented a part of the results focusing on the data-driven simulation.

In future studies, we will improve the proposed approach for even better design of air traffic management system. Hence, we will apply the various simulations with different conditions based on what-if analyses. Further, we will conduct Human-in-the-Loop simulations, which mimic actual radar operation in air traffic control, and evaluate the feasibility of the proposed arrival operation. Furthermore, we will consider operational constraints in the departure and surface operation at the airport (e.g., runway crossing, runway exit locations, taxiing routes and spot assignment), and further analyze the impacts of increasing air traffic on the runway capability and delay time. Extending our approach and combining data-driven analysis, theoretical evaluation, and simulation validation, the aim of this research will be to propose the best strategies for arrivals to work with departures and surface traffic in airport operation.

**Author Contributions:** Conceptualization, E.I.; methodology, E.I.; software, K.S. and K.K.; validation, K.S. and F.K.; formal analysis, K.S. and F.K.; investigation, K.S. and F.K.; resources, E.I. and K.K.; data curation, K.S. and F.K.; writing—original draft preparation, K.S., E.I., F.K. and K.K.; writing—review and editing, K.S., E.I. and K.K.; visualization, K.S. and F.K.; supervision, E.I. and K.K.; project administration, E.I.; funding acquisition, E.I. All authors have read and agreed to the published version of the manuscript.

**Funding:** This research was supported by JSPS KAKENHI Grant Number 20H04237.

**Institutional Review Board Statement:** Not applicable for studies not involving humans or animals.

**Informed Consent Statement:** Not applicable for studies not involving humans.

**Acknowledgments:** This research was conducted under CARATS initiatives supported by the Civil Aviation Bureau (JCAB) of the Japanese Ministry of Land, Infrastructure, Transport and Tourism as the "Studies on the Extended Arrival Management." It was also supported by JSPS KAKENHI Grant Number 20H04237. The authors are grateful to JCAB for providing air traffic data, and to Mayumi Ohnuki for assisting with data preparation and corrections to figures.

**Conflicts of Interest:** The funders had no role in the design of the study; in the collection, analyses, or interpretation of data; in the writing of the manuscript, or in the decision to publish the results.

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
