# Peer review of "Data-Driven Simulation for Evaluating the Impact of Lower Arrival Aircraft Separation on Available Airspace and Runway Capacity at Tokyo International Airport"

_aerospace, doi:10.3390/aerospace8060165_

Round 1

Reviewer 1 Report

Dear authors,

Thank you very much for your work. It was an interesting lecture. 

Some minor comments:

Lines 28-31: Line is too long, it is hard to understand.

Line 33: What do you by “…arrival traffic controlled equivalently to that…”?

Lines 36 – 56: Many different studies were analyzed by the authors to validate the importance of runway capacity. However, what were the findings of these studies? A brief one sentence explaining the results would be interesting. Similar as you did for your reference 15.

Line 86. Remove comma after configurations.

Line 87. Change “is conducted” for “were conducted”

Line 103 – 105. The new RECAT takes that into account Wing span instead of wing shape. Maybe you want to add wing shape to your listing.

Line 106. It might be a good idea to state that this separation is created in order to avoid catastrophic consequences. The reader not familiar with this might not understand the importance of this separation.

Line 112-113: “… according to the aircraft maximum….”. And remove “ of the aircraft”

Line 124. “Traffic volume is less”. Less than what? Do you mean that traffic levels are low ?

Line 145. Is there any reference for this 0.5% claim?

Line 219. Are you taking the more congested time as you are trying to estimate the maximum number of arrivals?

Line 223-223. It is not clear to me why you used a stochastic distribution.

Line 245. Yet another reason to add this 0.5% reference.

Figure 11 and Figure 12. Swap them. Figure 12 Is mentioned first in the text.

Table 4: Units (arrivals per hour) should be added.

Line 316: What is the reason behind the impact of these pop up flights?

Line 368. Why was BADA used in this work? .What were the BADA model parts relevant for you?

Line 379. Case 1 to 3 and Case 2 to 4 seem confusing. CASE 1 and 2 are ICAO and 3 and 4 are RECAT. Do you mean “by”?

Author Response

Reply to Reviewer’s Comments

Dear authors,

Thank you very much for your work. It was an interesting lecture.

>> We appreciate the reviewer’s important insights. In the following sections, you will find our responses to each of your points and suggestions. We are grateful for the time and energy you expended on our behalf.

Some minor comments:

1) Lines 28-31: Line is too long, it is hard to understand.

>> We separated line 28 – 31 into two sentences (see line 32 – 36 in the revised manuscript).

2) Line 33: What do you by “…arrival traffic controlled equivalently to that…”?

>> We deleted “…arrival traffic controlled equivalently to that…” to avoid the confusing explanation (see line 37 – 38 in the revised manuscript).

3) Lines 36 – 56: Many different studies were analyzed by the authors to validate the importance of runway capacity. However, what were the findings of these studies? A brief one sentence explaining the results would be interesting. Similar as you did for your reference 15.

>> We added the findings obtained in the literature [5], [7], [9], [11], [12], [13], [15] and [16] (see line 43 – 87 in the revised manuscript). Further, we added the description of developed methods in the literature [5], [7], [9] and [11], considering the other reviewer’s comment (see line 43 – 69 in the revised manuscript).

4) Line 86. Remove comma after configurations.

>> We removed comma as the reviewer had suggested (see line 117 in the revised manuscript).

5) Line 87. Change “is conducted” for “were conducted”

>> We changed “is conducted” for “were conducted” (see line 120 in the revised manuscript).

6) Line 103 – 105. The new RECAT takes that into account Wing span instead of wing shape. Maybe you want to add wing shape to your listing.

>> We changed “wing shape” for “wing span” (see line 135 in the revised manuscript).

7) Line 106. It might be a good idea to state that this separation is created in order to avoid catastrophic consequences. The reader not familiar with this might not understand the importance of this separation.

>> We added the sentence to demonstrate the idea that “the separation minima are created in order to avoid catastrophic consequences” (see line 136 in the revised manuscript).

8) Line 112-113: “… according to the aircraft maximum….”. And remove “ of the aircraft”

>> We remove “of the aircraft” (see line 148 in the revised manuscript).

9) Line 124. “Traffic volume is less”. Less than what? Do you mean that traffic levels are low ?

>> As the reviewer pointed out, we would like to tell traffic levels are low. Therefore, we changed “traffic volume is less” for “traffic volume is low” (see line 161 – 162 in the revised manuscript).

10) Line 145. Is there any reference for this 0.5% claim?

>> Unfortunately, the document is not open for publication because it describes the operational rules for air traffic controllers in Japan.

11) Line 219. Are you taking the more congested time as you are trying to estimate the maximum number of arrivals?

>> We utilized the flights arriving at Tokyo International Airport (RJTT) between 17:00 – 22:00. It is the most congested time period in nominal days (see line 260 in the revised manuscript).

12) Line 223-223. It is not clear to me why you used a stochastic distribution.

>> We used a stochastic distribution because we followed the procedure proposed in [31].

[31] Babak G. Jeddi, John F. Shortle, L.S. Statistical Separation Standards for the Aircraft Approach Process. Proc. 2006 IEEE/AIAA 25TH Digital Avionics Systems Conference 2006.

13) Line 245. Yet another reason to add this 0.5% reference.

>> As we replied to comment 10), this document is, unfortunately, not open for publication because it describes the operational rules for air traffic controllers in Japan.

14) Figure 11 and Figure 12. Swap them. Figure 12 Is mentioned first in the text.

>> We swapped Figure 11 and Figure 12 as the reviewer had suggested (see page 14 in the revised manuscript).

15) Table 4: Units (arrivals per hour) should be added.

>> We added the units (arrivals per hour) as the reviewer had suggested (see Table 4 in page 10 in the revised manuscript).

16) Line 316: What is the reason behind the impact of these pop up flights?

>> When these popup flights merge into the en-route air routes, the other cruising flights on the same routes should make spacing to avoid the conflict and keep safety separation among the popups. This adjustment increases the en-route and take-off delay for both of the popup and cruising flights; thus, the popup flights cause impacts on the air traffic flows. This adjustment could often occur when the departures from RJOO merge into the routes on which it is congested mainly due to the flights from RJFF, which hold the largest number of RJTT arrivals.

>> We added the reason behind the impact of the popup flights (see line 365 – 367 in the revised manuscript).

17) Line 368. Why was BADA used in this work? .What were the BADA model parts relevant for you?

>> We used the AirTOp simulator, which simulate the air traffic flow by using BADA model. The reason why we used BADA is because we would like to conduct the realistic simulation by incorporating the characteristics of aircraft type and corresponding speed at each flight phase (i.e., climb, cruise, and descent) into our simulations.

18) Line 379. Case 1 to 3 and Case 2 to 4 seem confusing. CASE 1 and 2 are ICAO and 3 and 4 are RECAT. Do you mean “by”?

>> We appreciate the reviewer’s kind comment. We would like to tell ICAO standard is replaced “by” RECAT as the reviewer pointed out. Therefore, we added the sentence “Case 1 and 2 adopted ICAO standard while Case 3 and 4 utilized RECAT” (see line 430 in the revised manuscript) and “When ICAO standard is replaced by RECAT” (see line 431 – 432 in the revised manuscript).

>> Again, we appreciate all of your insightful comments. We worked hard to be responsive to them. Thank you for taking the time and energy to help us improve the paper.

Reviewer 2 Report

The topic of the paper is to Data-driven Simulation for Evaluating the Impact of Lower Arrival Aircraft Separation on Available Runway and Airspace Capacity, which should be more specific, not a general introduction.  

The presentation of the research method is unclear, they often refer to literature, scientific research by the authors of e.g. Bubalo, Irvine, Tamas, what is their own scientific and research contribution? only data simulation and comparison? Please give me more details about your work.

The results with the description of the simulation environment are rather incomplete (presentation of the method, simulation model), more details should be provided from the developed methods and theoretical results of the authors Bubalo, Irvine, Tamas.

This is a scientific article, the authors refer the reader to chapters, tables and figures throughout the article. This is not a test report .. see pt. 1 Introduction, 79-94., See pt. 4, 213, 214, 4.1, 72, etc.

In conclusion, reference should be made to the interesting results obtained and their simulations, supported by the results of the research conducted by the authors.

What scientific and functional outcomes have been achieved.

Please follow the instructions for manuscript preparation.

Language, manuscript text.

The presented article is descriptive rather than scientific. I hope that the final version will be able to show more simulations and results of the authors.

Sugestions:

  1. Please follow the directions for preparation of the manuscript. Language of manuscript text, bibliography - references, see page 5, what is the correct RECAT acronym, see Abstract, Keywords and Introduction.
  2. This is a scientific article, the authors refer the reader to chapters, tables and figures throughout the article. This is not a test report .. see pt. 1 Introduction, 79-94., See pt. 4, 213, 214, 4.1, 72, etc.
  3. Correct the description to figure 4, markings (a), (b).
  4. Correct the description to Figures 1, 2, 3, these are not definitions !!!
  5. Insert picture 11, 12, 13 with a different color, because it is not legible.
  6. The summary should refer to the interesting results obtained and be significantly detailed and extensive.
  7. What new and important aspects the work brings to learning.
  8. What scientific and application effects have been achieved.

Author Response

Reply to Reviewer’s Comments

>> We appreciate the reviewer’s important insights. In the following sections, you will find our responses to all suggestions each by each. We are grateful for the time and energy you expended on our behalf.

1) The topic of the paper is to Data-driven Simulation for Evaluating the Impact of Lower Arrival Aircraft Separation on Available Runway and Airspace Capacity, which should be more specific, not a general introduction.

>> We appreciate the reviewer’s suggestion. As the reviewer had recommended, we changed the title for this paper to “Data-driven Simulation for Evaluating the Impact of Lower Arrival Aircraft Separation on Available Airspace and Runway Capacity at Tokyo International Airport”. We added the word, “Tokyo International Airport”, to take into account the characteristic of the region (see the title of the revised manuscript).

2) The presentation of the research method is unclear, they often refer to literature, scientific research by the authors of e.g. Bubalo, Irvine, Tamas, what is their own scientific and research contribution? only data simulation and comparison? Please give me more details about your work.

>> Our main challenge is to propose a scientific and systematic way of designing future air traffic management. As we summarized in paragraph 3 in chapter 1 (line 97 – 109 in the revised manuscript), firstly we theoretically analyzed the optimal balance between increasing arrival rate in the airspace based on two types of queuing models. These theoretical studies contributed to propose design requirements of arrival management system at Tokyo International Airport, but it could not consider detailed characteristics of aircraft behaviors and operational rules in the airspace and airport. Then, we validated the theoretical results based on the data-driven simulation method. In addition, this paper clarified available runway capacity using stochastic data analysis. To summarize, this study contributed to build a systematic approach integrating data science, theoretical modeling, and simulation evaluation for designing future air traffic management. This paper presented a part of our study focusing stochastic data analytics and simulation evaluation. In the third paragraph in the introduction chapter (line 97 – 109 in the revised manuscript), we explained the scientific and research contribution in this paper.

>> For further clarifying the reviewer’s question, we summarized the contribution of this study in chapter 8 (see line 525 – 528 in the revised manuscript).

3) The results with the description of the simulation environment are rather incomplete (presentation of the method, simulation model), more details should be provided from the developed methods and theoretical results of the authors Bubalo, Irvine, Tamas.

>> We appreciate the reviewer’s insightful comment. We summarized more details on the developed methods and theoretical results of the Bubalo, Irvine, Tamas (see line 41 – 69 in the revised manuscript).

4) This is a scientific article, the authors refer the reader to chapters, tables and figures throughout the article. This is not a test report .. see pt. 1 Introduction, 79-94., See pt. 4, 213, 214, 4.1, 72, etc.

>> We changed “section” for “chapter”. Further, we referred all figures and tables throughout the revised manuscript.

5) In conclusion, reference should be made to the interesting results obtained and their simulations, supported by the results of the research conducted by the authors.

>> We summarized new findings and references in chapter 7, discussion chapter (see line 499 – 515 in the revised manuscript). Further, we stated the interesting results in chapter 8 as we followed the reviewer’s comment (see line 521 – 523 in the revised manuscript).

6) What scientific and functional outcomes have been achieved.

>> We appreciate the reviewer’s comment. Please refer the reply to comment 2).

Please follow the instructions for manuscript preparation.

Language, manuscript text.

7) The presented article is descriptive rather than scientific. I hope that the final version will be able to show more simulations and results of the authors.

>> We appreciate the reviewer’s kind comment. We ran various simulation scenarios with different conditions, but the volume of results was larger than one paper. We will introduce more case studies with what-if analyses in the future publications. We summarized our future works in chapter 8 (see line 530 – 531 in the revised manuscript).

Sugestions:

8) Please follow the directions for preparation of the manuscript. Language of manuscript text, bibliography - references, see page 5, what is the correct RECAT acronym, see Abstract, Keywords and Introduction.

>> RECAT stands for “wake turbulence categories RE-CATegorization”. We unified the same terminology throughout the paper (in Abstract, Keywords, and Introduction).

9) This is a scientific article, the authors refer the reader to chapters, tables and figures throughout the article. This is not a test report .. see pt. 1 Introduction, 79-94., See pt. 4, 213, 214, 4.1, 72, etc.

>> We changed “section” for “chapter”. Further, we referred all figures and tables throughout the revised manuscript.

10) Correct the description to figure 4, markings (a), (b).

>> We revised the caption of Figure 4(a) “Northerly wind operation” (see page 7 in the revised manuscript).

11) Correct the description to Figures 1, 2, 3, these are not definitions !!!

>> We erased “Definition of ” from the caption of Figure 1 to 3 (see page 4 and 6 in the revised manuscript).

12) Insert picture 11, 12, 13 with a different color, because it is not legible.

>> We inserted the Figure 11 – 13 with a different color (see page 14 – 15 in the revised manuscript).

13) The summary should refer to the interesting results obtained and be significantly detailed and extensive.

>> We summarized the interesting results in chapter 7 (see line 499 – 515 in the revised manuscript), and also briefly summarized in the concluding remarks (see line 521 – 523 in the revised manuscript). Further, we will introduce various simulation results with more case studies and conduct what-if analyses in the future work. We also summarized the future works in chapter 8 (see line 530 – 531 in the revised manuscript).

14) What new and important aspects the work brings to learning.

>> We appreciate the reviewer’s comment. Please refer to the reply comment 2).

15) What scientific and application effects have been achieved.

>> We appreciate the reviewer’s comment. Please refer to the reply comment 2).

>> Again, we appreciate all of your insightful comments. We worked hard to be responsive to them. Thank you for taking the time and energy to help us improve the paper.

Reviewer 3 Report

This paper provides a validating work on improvements in total delay time by reducing the arrival aircraft separation. In general, the presentation is clear and easy to follow. Here are some comments on this work.  

Firstly, the significance of the obtained results is not well elaborated. The improvements in total delay time are achieved by implementing reduced separation and with the assumption that the runway capacity is not an issue. Introducing more case studies with what-if analyses might be helpful to obtain more insightful results.  

Secondly, please clarify some observations in the simulation studies. 1) it seems that results from RECAT have a longer tail than results from ICAO, which would be the reason (in Fig. 14); 2) outliers in Fig. 16 are significant. It would be better if the authors could clarify more on this. Would it be some abnormal scenarios introduced by the simulator itself?  

Thirdly, if the FCFS rule is not a constraint, would it be further improved on the performance?  

In general, this paper presents interesting results and validations on the RECAT concept and the reviewer would eager to see more results under varying conditions.  

Author Response

Reply to Reviewer’s Comments

This paper provides a validating work on improvements in total delay time by reducing the arrival aircraft separation. In general, the presentation is clear and easy to follow. Here are some comments on this work. 

>> We appreciate the reviewer’s important insights. In the following sections, you will find our responses to each of your points and suggestions. We are grateful for the time and energy you expended on our behalf.

1) Firstly, the significance of the obtained results is not well elaborated. The improvements in total delay time are achieved by implementing reduced separation and with the assumption that the runway capacity is not an issue. Introducing more case studies with what-if analyses might be helpful to obtain more insightful results.

>> Our main challenge is to propose a scientific and systematic way of designing future air traffic management. As we summarized in paragraph 3 in chapter 1 (line 97 – 109 in the revised manuscript), we evaluated the theoretical results based on the data-driven simulation evaluation in this paper. In addition, this paper clarified available runway capacity using stochastic data analysis.

The significance of the obtained results in this paper is the quantitative validation of the theoretical results based on the data-driven simulation evaluation. The contributions are summarized in chapter 8 (see line 5in the revised manuscript). Definitely, we will introduce more case studies with what-if analyses in the future work. We summarized the future works in chapter 8 (see line 530 – 531 in the revised manuscript).

2) Secondly, please clarify some observations in the simulation studies. 1) it seems that results from RECAT have a longer tail than results from ICAO, which would be the reason (in Fig. 14); 2) outliers in Fig. 16 are significant. It would be better if the authors could clarify more on this. Would it be some abnormal scenarios introduced by the simulator itself? 

>> With regard to the first comment, the tail length of both ICAO standard and RECAT distributions are almost the same. It seems that the RECAT results have a longer tail than ICAO’s because these two tail distributions are overlapped. With regard to the second comment, the outliers were significant because the airspace capacity in Case 1 and 3 is 4 [ac/hr] less than that in Case 2 and 4. The arrival traffic flows in Case 1 and 3 were controlled to keep the assigned airspace capacity. Therefore, the results showed that some flights arriving at RJTT during the congested time were delayed at the gate (i.e., at the departure airport).

>> We explained the overlap of the tail distribution (see line 438 – 440 in the revised manuscript).

3) Thirdly, if the FCFS rule is not a constraint, would it be further improved on the performance?

>> In our opinion, the performance would not be improved more than FCFS rule. Optimizing the sequence should be carefully discussed in order not to impact on the entire air traffic flow. If the aircraft would be holding in the air for the purpose of the position shifting, it increases ATCos workloads, and the delay time may propagate to the succeeding air traffic. Furthermore, it is not a fair handling for the airlines. One relevant reason for the position shifting is to increase the runway throughput. However, it was not the case at RJTT because applying the FCFS rule was enough to satisfy the maximum runway capacity (41 [ac/hr] at RWY34L), which was estimated in this study. Therefore, as discussed in chapter 7 (Discussion), FCFS rule would be enough unless the demand for a runway exceeds the estimated maximum runway capacity.

4) In general, this paper presents interesting results and validations on the RECAT concept and the reviewer would eager to see more results under varying conditions.

>> We appreciate the reviewer’s kind comment. Following the reviewer’s suggestion, we will conduct the various simulations with different conditions in the future. We added the future work in chapter 8 (see line 530 – 531 in the revised manuscript).

>> Again, we appreciate all of your insightful comments. We worked hard to be responsive to them. Thank you for taking the time and energy to help us improve the paper.

Round 2

Reviewer 2 Report

Thank you to the authors for addressing my comments and suggestions on an interesting article. Accept in the present form,

kind regards 

Reviewer 3 Report

Glad to see all the amendments made to improve the quality. I've no further comment on this work.